# Graph Information Matters: Understanding Graph Filters from Interaction Probability

## Abstract

Graph Neural Networks (GNNs) have received extensive affirmation for their promising performance in graph learning problems. Despite their various neural architectures, most are intrinsically graph filters that provide theoretical foundations for model explanations. In particular, low-pass filters show superiority in label prediction in many benchmarks. However, recent empirical research suggests that models with only low-pass filters do not always perform well. Although increasing attempts to understand graph filters, it is unclear how a particular graph affects the performance of different filters. In this paper, we carry out a comprehensive theoretical analysis of the synergy of graph structure and node features on graph filters' behaviors in node classification, relying on the introduction of interaction probability and frequency distribution. We show that the homophily degree of graphs significantly affects the prediction error of graph filters. Our theory provides a guideline for graph filters design in a data-driven manner. Since it is hard for a single graph filter to live up to this, we propose a general strategy for exploring a data-specified filter bank. Experimental results show that our model achieves consistent and significant performance improvements across all benchmarks. Furthermore, we empirically validate our theoretical analysis and explain the behavior of baselines and our model.

## 1 Introduction

Graph Neural Networks (GNNs) have continuously attracted interest as their promising performance in various graph learning problems. It is known that most of GNNs are intrinsically graph filters (Kipf & Welling, 2017; Defferrard et al., 2016; Ortega et al., 2018; Nt & Maehara, 2019). With the theoretical foundation of filters, there is an increasing attempt at model explanation, e.g. explaining the behavior of various GNNs in node classification. Nt & Maehara (2019) investigated the superiority of low-pass filters backed up with theoretical arguments while recent research (Balcilar et al., 2020; Chang et al., 2020; Bo et al., 2021) empirically revealed the weakness of GNNs with only low-pass filters in certain datasets. These contradictory views on low-pass filters pose a significant problem: Why does a filter work on one dataset but not on another? More precisely, for a given filter, what kinds of structure and features are useful for prediction? This makes it clear to us that in order to solve this problem, it is necessary to take into account graph information, including the graph structure, features, and labels.

Existing theoretical research is mostly restricted to the investigation of filters themselves such as exploring their expressive power (Oono & Suzuki, 2020; Balcilar et al., 2020), without considering their inconsistency of performance on different graphs. It is clear that structural and feature information lead to the possible inconsistency. However, there has been little explicit analysis of how graph information influences the performance of graph filters. For instance, GNNs have formulated a variety of graph filters in a heuristic manner under a suppressed homophily assumption, i.e., nodes with similar attributes/labels tend to have connections. There remains a paucity of quantitative description of homophily until Pei et al. (2020) designed a rough index to measure it.

In this paper, we establish a comprehensive theoretical analysis of the effect of structure and feature information on node label prediction to fill the gap and provide deep insights into the explanation of graph filters. We first establish a systematic investigation on graphs with an indicator in terms of homophily - the *interaction probability* and a distributional representation of input information - the

*frequency distribution*. The interaction probability derived from random walk theory relates node labels with its local topology and quantifies the degree of clustering of nodes in the same/different class. We argue that interaction probability reflects the difficulty in identifying one class from others. In terms of feature information, we draw on spectral analysis representing features as frequency distributions. Furthermore, we consider the moment of frequency and build an explicit relation with graph structure. Interestingly, we find that the moment of label frequency (noting that a one-hot label vector can be regarded as a special node feature) is determined by interaction probability. The aforementioned preparations underpin our deep understanding of graph filters.

We validate the prediction error of a graph filter under two settings: a. *fixed graph structure, unravel the influence of input (original or transformed node features)*; b. *given input, show how structure matters*, and provide analysis utilizing frequency distribution and interaction probability. The main conclusions are: 1. given structure, the frequency response of an ideal graph filter should be consistent with the main frequency band of label frequency, that is, a matched frequency response is the premise of success; 2. given input, a graph filter essentially tunes the weight of edges - failing to make a homophily degree large enough may cause an unsatisfactory prediction accuracy. These interpretations of graph filters imply a data-driven filter design principle. In addition, we apply these theoretical results to three types of filters - low-pass, high-pass, and band-pass filters with specified form. It shows that a single graph filter is hard to comply with the principle of ideal filters, especially when the homophily degree and label frequency distribution of different classes are very different. For example, when frequency distributions of labels are far from each other, it is hard to find a single filter whose frequency response can cover all the main frequency bands well. In this paper, we leverage a combination of band-pass graph filters to overcome this problem and develop a simple yet effective framework to show how to learn multiple filters depending on datasets. We empirically validate our theoretical analysis and investigate structure and feature information of benchmarks. We verify our model on a variety of datasets and explain the behavior of baselines and our model. Experimental results show that our model achieves a consistent and significant performance improvement across all benchmarks.

Our main contributions are: 1.We develop a theoretical analysis of graph information based on the introduction of *interaction probability* and *frequency distribution*; 2.We provide a deep understanding of the performance of graph filters illustrating how graph structure and input information matter; 3.We indicate the weakness of GNNs with a single graph filter and propose a general framework to learn a *data-specified filter bank* which contributes to significant improvement.

## 2  RELATED WORK

In this paper, we focus on the analysis of graph filters in the context of graph neural networks. Since Bruna et al. (2014) defined spectral graph filters and extended convolutional operations to graphs, various spectral graph neural networks have been developed. For example, ChebNet (Defferrard et al., 2016) defines the Chebyshev polynomial filter which can be exactly localized in the k-hop neighborhood. Kipf & Welling (2017) simplified the Chebyshev filters using a first-order approximation and derived the well-known graph convolutional networks (GCNs). Bianchi et al. (2021) proposed the rational auto-regressive moving average graph filters (ARMA) which are more powerful in modeling the localization and provide more flexible graph frequency response, however more computationally expensive and also more unstable. Very recently, Min et al. (2020) augmented conventional GCNs with geometric scattering transforms which enabled band-pass filtering of graph signals and alleviated the oversmoothing issue. In addition, most graph neural networks originally defined in the spatial domain are also found essentially connected to the spectral filtering (Balcilar et al., 2020). By bridging the gap between spatial and spectral graph neural networks, Balcilar et al. (2020) further investigated the expressiveness of all graph neural networks from their spectral analysis. However, their analysis is limited to the spectrum coverage of a graph filter itself and lacks deeper insights into the graph-dependent performance of these filters.

Another related topic is the measurement of graph homophily. Beyond the interaction probability that we define in this paper, there are some other heuristic metrics for homophily. Pei et al. (2020) defined a node homophily index to characterize their datasets and help explain their experimental results for Geom_GCN: $\beta = \frac{1}{\#\text{nodes}} \sum_v \frac{\#\text{neighbors of } v \text{ that have the same label as } v}{\#\text{neighbors of } v}$. Zhu et al. (2020) defined edge homophily ratio instead and identified a set of key designs that can boost learning from the graph

structure in heterophily: $h = \frac{\#\text{edges whose end nodes have same labels}}{\#\text{edge}}$. This edge homophily definition is sensitive to the number of classes and size of each class, and Lim et al. (2021) made a modification to alleviate this problem. Our work differentiates from these works in that we not only use our definition to characterize the graph but also directly relate it to the performance of graph filters (or GNNs).

# 3 THEORETICAL ANALYSIS OF GRAPH INFORMATION

## 3.1 NOTATION

Let $\mathcal{G}_n = (\mathcal{V}_n, \mathcal{E}_n)$ be an undirected graph with additional self-connection, where $\mathcal{V}_n = \{v_0, \ldots, v_{n-1}\}$ is the set of nodes and $\mathcal{E}_n \subset \mathcal{V}_n \times \mathcal{V}_n$ is the set of edges. Let $A \in \mathbb{R}^{n \times n}$ be the adjacency matrix and $L = D - A$ be the Laplacian matrix, where $D$ is a diagonal degree matrix with $D_{ii} = \sum_j A_{ij}$. we denote $\tilde{A} = D^{-\frac{1}{2}} A D^{-\frac{1}{2}}$, then $\tilde{L} = D^{-\frac{1}{2}} L D^{-\frac{1}{2}} = I - \tilde{A}$ is the symmetric normalized Laplacian. Let $(\lambda_i, \mathbf{u}_i)$ be a pair of eigenvalue and unit eigenvector of $\tilde{L}$, where $0 = \lambda_0 \leq \cdots \leq \lambda_{n-1} \leq 2$.

## 3.2 PROBLEM SETTING

In this paper, we are mainly interested in node classification problems on undirected graphs. Given $\mathcal{G}_n = (\mathcal{V}_n, \mathcal{E}_n)$, we consider $\mathcal{T} = \{0, \ldots, K-1\}$ as the set of all node labels. For $\forall k \in \mathcal{T}$, we denote $\mathcal{C}_k$ as the set of nodes with label $k$ and $R \in \mathbb{R}^{K \times K}$ as a size matrix which is a diagonal matrix with $R_k = |\mathcal{C}_k|$. Considering single-label problems in which classes are mutually exclusive, we use one-hot encoding to indicate the class label and introduce a label matrix $Y \in \mathbb{R}^{n \times K} = (\mathbf{y}_0, \ldots, \mathbf{y}_{K-1})$ to represent the labels of $\mathcal{V}_n$, where $\mathbf{y}_k$ is the indicator vector of $\mathcal{C}_k$. Obviously, $R = Y^\top Y$, $Y^\top \mathbb{1} = diag(R)$ and $Y \mathbb{1} = \mathbb{1}$. A signal $\mathbf{x}$ on $\mathcal{G}_n$ can be arranged the signal values in a vector form $\mathbf{x} = (x_0, \ldots, x_{n-1})^\top$. Particularly, labels $\{\mathbf{y}_k | k \in \mathcal{T}\}$ are also graph signals.

## 3.3 A STRUCTURE INDICATOR - INTERACTION PROBABILITY

Homophily of graphs is an implicit assumption widely leveraged in graph learning methods including GNNs. It is considered an indisputable common property of most graphs, despite its descriptive and unquantifiable definition, which introduces a variety of uncertainties. In this section, starting with the random walk, we introduce interaction probability to overcome this challenge.

For a random walk on $\mathcal{G}_n$, we denote $P = D^{-1} A$ as its transition matrix which is also a row Markov matrix. From the random walk theory, $P^k$ is the $k$-step transition matrix, and $P_{ij}^k$ is the probability that a random walker starting from node $v_i$ arrives at $v_j$ after $k$ steps. For a node $v$ and a class $\mathcal{C}_l$, we denote $\pi_i^k(\mathcal{C}_l)$ as the probability that a random walker starting from $v_i$ stays in $\mathcal{C}_l$ at the $k$-th step. It is trivial that $\pi_i^k(\mathcal{C}_l) = \sum_{j \in \mathcal{C}_l} P_{ij}^k$ with $\sum_{l \in \mathcal{T}} \pi_i^k(\mathcal{C}_l) = 1$. $\pi_i^k(\mathcal{C}_l)$ demonstrates the relative preference/closeness of node $v_i$ for $\mathcal{C}_l$ with $k$-scale. To meet the homophily assumption, for $v_i$ in $\mathcal{C}_l$, $\pi_i^k(\mathcal{C}_l)$ is expected to gap away from others. Since $\pi_i^k(\mathcal{C}_l) - \sum_{m \neq l} \pi_i^k(\mathcal{C}_m) = 2\pi_i^k(\mathcal{C}_l) - 1$, $\pi_i^k(\mathcal{C}_l)$ can be regarded as a measure of the $k$-scale homophily degree of node $v_i$. Particularly, for $\forall k \in \mathbb{N}$ and $v_i \in \mathcal{C}_l$, $\pi_i^k(\mathcal{C}_l) = 1$ means that $\mathcal{C}_l$ is a community and will never communicate with other classes. However, this case is rare in real graphs. Below, we investigate the homophily of a class and propose a method to measure the communication strength between two classes.

**Definition 3.1** (*k*-step interaction probability). For $l, m \in \mathcal{T}$, we define $\Pi^k$ as the $k$-step interaction probability matrix formulated as follows:

$$\Pi_{lm}^k = \frac{1}{R_l} \sum_{v_i \in \mathcal{C}_l} \pi_i^k(\mathcal{C}_m) = \frac{1}{R_l} \sum_{v_i \in \mathcal{C}_l, v_j \in \mathcal{C}_m} P_{ij}^k = \frac{\mathbf{y}_l^\top P^k \mathbf{y}_m}{\mathbf{y}_l^\top \mathbf{y}_l} \tag{1}$$

$$\Pi^k = (Y^\top Y)^{-1} Y^\top P^k Y = R^{-1} Y^\top P^k Y. \tag{2}$$

$\Pi_{lm}^k$ is the probability that a random walker from $\mathcal{C}_l$ arrives at $\mathcal{C}_m$ after $k$ steps.

**Remark 1.** Obviously, $\Pi^k \mathbb{1} = \mathbb{1}$. $\Pi_{lm}^k$ is the mean proportion of $\mathcal{C}_m$ in the $k$-hop neighbors of nodes from $\mathcal{C}_l$. Noting that $\text{rank}(Y) = K$, when $K \neq n$, $Y R^{-1} Y^\top \neq I$, thus $(R^{-1} Y^\top P Y)^k \neq R^{-1} Y^\top P^k Y$, i.e. $(\Pi)^m \neq \Pi^m$. More generally, for an arbitrary polynomial function $g$, $R^{-1} Y^\top g(P) Y$ is likely not equal to $g(R^{-1} Y^\top P^k Y)$. In the rest of paper, we

write $\tilde{g}(\Pi) = R^{-1}Y^\top g(P)Y$ and $g(\Pi) = g(R^{-1}Y^\top P^k Y)$. For instance, if $g(\cdot) = (\cdot)^m$, then $\tilde{g}(\Pi) = \Pi^m$ and $g(\Pi) = (\Pi)^m$. Also, we denote $\Pi_{ll}^k$, the *self-interaction probability*, as $\pi_l^k$ for short.

1-step interaction probability intuitively reflects the degree of clustering of two classes and $\sum_{i=1}^k \Pi^i$ measures the strength of interaction between classes in the scale of $k$ steps. Since $P$ is not symmetric, $\Pi_{lm}^k \neq \Pi_{ml}^k$. To facilitate analysis, here we propose *a symmetric variant of interaction probability* to identify the interactions between two classes. We denote this symmetric $k$-step interaction probability matrix as $\tilde{\Pi}^k$, by replacing $P$ with $\tilde{A} = D^{-\frac{1}{2}}AD^{-\frac{1}{2}}$, we obtain $\tilde{\Pi}^k = R^{-\frac{1}{2}}Y^\top \tilde{A}^k Y R^{-\frac{1}{2}}$. Below, we investigate the important properties of $\Pi^k$ and $\tilde{\Pi}^k$.

**Proposition 3.1.** For $l, m \in \mathcal{T}$ and an arbitrary polynomial function $g(\cdot)$, we have:
a. $R_l \Pi_{lm}^k + R_m \Pi_{ml}^k \geq 2\sqrt{R_l R_m}\tilde{\Pi}_{lm}^k$, where $R_l$ is the $l$-th diagonal element of $R$;
b. $(\tilde{g}^2(\tilde{\Pi}))_{ll} \geq (\tilde{g}(\tilde{\Pi})_{ll})^2$, where $\tilde{g}^k(\tilde{\Pi}) = R^{-\frac{1}{2}}Y^\top g^k(\tilde{A})Y R^{-\frac{1}{2}}$.

The proof can be found in Appendix B. Since $\tilde{\Pi}^k \mathbb{1} \neq \mathbb{1}$, that is, the measure is no longer a probability measure. However, according to Prop.3.1.a ( let $m = l$ ), $\tilde{\pi}_l^k$ is the lower bound of $\pi_l^k$, and $\tilde{\pi}_l^k = \pi_l^k$ when $\mathcal{G}_n$ is a regular graph. In the rest of theoretical analysis, we use $\tilde{\pi}_l^k$ to measure the degree of $\mathcal{C}_l$'s clustering. Let $g(\cdot) = (\cdot)^k$, from Prop.3.1.b, we have $\tilde{\pi}_l^{2k} \geq (\tilde{\pi}_l^k)^2$. In Section.4.2, we leverage this inequality to derive a lower bound of our prediction error and further illustrate how structure influences the performance of a given filter.

## 3.4 A Feature Indicator - Frequency Distribution

Following the graph signal processing (GSP) concepts, $\lambda_0, \ldots, \lambda_{n-1}$ are graph frequencies and $\mathbf{u}_0 \ldots, \mathbf{u}_{n-1}$ are the corresponding frequency components which are invariant of graph filters. Through Fourier transform, we obtain $\{\alpha_i = \langle \mathbf{u}_i, \mathbf{x}\rangle | i = 0, \ldots, n-1\}$ the spectral representation of a graph signal $\mathbf{x}$, called graph signal spectrum. Moreover, a graph signal can be represented as a linear combination of frequency components, i.e., $\mathbf{x} = \sum \alpha_i \mathbf{u}_i$. For a label vector $\mathbf{y}_l$ which is also a graph signal, we denote $\{\gamma_0, \ldots, \gamma_{n-1}\}$ as its spectrum. There is an intuitive assumption: *information of label vectors is all we need for classification* - we will validate this assumption in Section 4.1. Under this context, $\gamma_i^2 / \sum_i \gamma_i^2$ reflects how much the frequency component $\mathbf{u}_k$ contributes to the distinctiveness of $\mathcal{C}_l$, without considering the positivity and negativity of effects. Interestingly, we find that the normalized signal spectrum is a histogram/discrete distribution defined below.

**Definition 3.2** (Frequency distribution). We define $\mathbf{f}$, the frequency of signal $\mathbf{x}$, as a random variable taking values in the set of graph frequencies with probability $\Pr(\mathbf{f} = \lambda_k) = \alpha_k^2 / \sum_i \alpha_i^2$. The probability describes the frequency distribution of signal $\mathbf{x}$.

With this definition, we derive distributional representations of signals from their spectral representations/spectra. One can evaluate the signal effect by comparing frequency distributions of signals and label vectors under a specified distribution metric, such as Wasserstein distance. Below, we consider the moment of frequency distribution to show how graph structure influences signal frequency.

**Proposition 3.2.** For $\mathcal{G} = \{\mathcal{V}, \mathcal{E}\}$, let $\mathbf{f}$ be the frequency of signal $\mathbf{x}$, then $\mathbb{E}[\mathbf{f}^n] = \frac{\mathbf{x}^\top (I - \tilde{A})^n \mathbf{x}}{\mathbf{x}^\top \mathbf{x}}$.

The proof of this proposition can be found in Appendix B. With the definition of interaction probability, we further represent the moment of the label vector's frequency.

**Corollary 3.3.** For label frequency $\mathbf{f}_l$ of $\mathbf{y}_l$, we have $\mathbb{E}[\mathbf{f}_l^n] = (\tilde{g}(I - \tilde{\Pi}))_{ll}$ with $g = (\cdot)^n$.

Recall that $\tilde{g}(I - \tilde{\Pi}) = R^{-\frac{1}{2}}Y^\top (I - \tilde{A})^n Y R^{-\frac{1}{2}}$, we have $\mathbb{E}[\mathbf{f}_l] = 1 - \tilde{\pi}_l$, $\mathbb{E}[\mathbf{f}_l^2] = 1 - 2\tilde{\pi}_l + \tilde{\pi}_l^2$ and the variance of $\mathbf{f}_l$: $\text{Var}(\mathbf{f}_l) = \tilde{\pi}_l^2 - (\tilde{\pi}_l)^2$. It can be seen that both the mean and variance of label frequency are close to 0 when $\tilde{\pi}_l$ approaches 1, which reflects a high homophily degree (as $\tilde{\pi}_l \leq \pi_l \leq 1$). In Section 4.1, we conduct a more detailed analysis of feature information of spectral space with frequency distribution.

## 4 Analysis of Graph Filters

A graph filter is defined as a function $g$ with applied Laplacian matrix or adjacency matrix. Denote $\mathbb{R}[\tilde{A}]$ as a polynomial ring in $\tilde{A}$ over $\mathbb{R}$, here we are mainly interested in $g \in \mathbb{R}[\tilde{A}]$. In this section, we

provide a deep understanding of the performance of graph filters concerning label prediction based on the above theoretical analysis of graph information. In general, there are two major concerns: *with fixed graph structure, how does the input impact the performance of a given filter?* and *with fixed input, how does graph structure impact the performance of a given filter?*. In this section, we provide the theoretical analysis of these two questions in Sections 4.1 and 4.2, respectively.

The general formulation of the $l + 1$-th layer of spectral GNNs is $X^{(l+1)} = \sigma(g(\tilde{A})X^{(l)}W^{(l+1)})$, here $\sigma$ is an activation function, $X^{(l)}$ is the output of the $l$-th layer, $X^{(0)}$ is a feature matrix and $W^{(l+1)}$ is a learnable transformation matrix. We call $X^{(l)}W^{(l+1)}$ the input of $g(\tilde{A})$ in $l + 1$-th layer and denote $X$ as the input of $g(\tilde{A})$ in the last layer. In the following sections, we discuss the prediction error of spectral GNNs with a given graph filter without activation function before prediction. That is, in the last layer with $X$ as input, $g(\tilde{A})X$ is directly used for prediction.

**Definition 4.1** (Prediction error). Let $X \in \mathbb{R}^{n \times K}$ be the input of graph filter $g(\tilde{A})$, $Y \in \mathbb{R}^{n \times K}$ is the label matrix, the prediction error is formulated by:

$$Er(g, X) = \| g(\tilde{A})X - Y \|_F^2 = \text{tr}(X^\top g^2(\tilde{A})X) - 2\text{tr}(X^\top g(\tilde{A})Y) + \| Y \|_F^2 \qquad (3)$$

**Remark 2.** For a label vector $\mathbf{y}_l$, we denote $Er(g, \mathbf{x}_l) = \| g(\tilde{A})\mathbf{x}_l - \mathbf{y}_l \|_F^2$ as the error of $g(\tilde{A})$ predicting class $l$. Obviously, $Er(g, X) = \sum_{l \in \mathcal{T}} Er(g, \mathbf{x}_l)$, where $\mathbf{x}_l$ is the $l$-th column of $X$.

In particular, we will apply our conclusion to specified filters and make concrete analysis.

**Definition 4.2.** With $\epsilon \in [0, \epsilon_0]$ and $\epsilon' \in [-1, 1]$, $\epsilon_0$ is a small constant, we define low-pass filters $g_{l(\epsilon)}(\tilde{A})$, high-pass filters $g_{h(\epsilon)}(\tilde{A})$ and band-pass filters $g_{b(\epsilon')}(\tilde{A})$ as:

$$g_{l(\epsilon)}(\tilde{A}) = \epsilon I + \tilde{A}, \quad g_{h(\epsilon)}(\tilde{A}) = \epsilon I - \tilde{A}, \quad g_{b(\epsilon')}(\tilde{A}) = I - (1 + |\epsilon'|)^{-2}(\epsilon' I - \tilde{A})^2.$$

For $\lambda$, an eigenvalue of $\tilde{L}$, we have $g_{l(\epsilon)}(\lambda) \in [\epsilon - 1, 1 + \epsilon]$, $g_{h(\epsilon)}(\lambda) \in [\epsilon - 1, 1 + \epsilon]$ and $g_{b(\epsilon')}(\lambda) \in [0, 1]$ since $\lambda \in [0, 2]$. Particularly, $g_{l(0)}$ is the GCN filter.

## 4.1 How Input Matters

Denote $\tilde{X} = U^\top X = (\tilde{\mathbf{x}}_0, \ldots, \tilde{\mathbf{x}}_{K-1})$ and $\tilde{Y} = U^\top Y = (\tilde{\mathbf{y}}_0, \ldots, \tilde{\mathbf{y}}_{K-1})$, where $U$ is a matrix with unit eigenvectors of $\tilde{L}$ (recall that eigenvectors of $\tilde{A}$ are consistent with that of $\tilde{L}$), revisiting $Er(g, \mathbf{x}_l)$ and $Er(g, \mathbf{y}_l)$ in spectral domain, we have:

$$Er(g, \mathbf{x}_l) = \| g(I - \Lambda)\tilde{\mathbf{x}}_l - \tilde{\mathbf{y}}_l \|_F^2 = \sum_i (g(1 - \lambda_i)\alpha_i - \gamma_i)^2 \qquad (4)$$

$$Er(g, \mathbf{y}_l) = \sum_i \gamma^2 (1 - g(1 - \lambda_i))^2 = R_l \sum_i p_i (1 - g(1 - \lambda_i))^2 = R_l \mathbb{E}[1 - g(1 - \mathbf{f}_l)]^2 \qquad (5)$$

where $\Lambda$ is the eigenvalue matrix of $\tilde{L}$, $\alpha_i$ and $\gamma_i$ are the spectra of $\mathbf{x}_l$ and $\mathbf{y}_l$ respectively, $p_i = \Pr(\mathbf{f}_l = \lambda_i)$, $\mathbf{f}_l$ is the frequency of $\mathbf{y}_l$. For better comparison, we normalize the input $\mathbf{x}_l$: $\| \mathbf{x}_l \|_F^2 = \| \mathbf{y}_l \|_F^2$, i.e., $\sum \alpha_i^2 = \sum \gamma_i^2$. $g$ is re-scaled function with $g([0, 2])$ concentrating in $[-1, 1]$.

**How input information matter?** With normalized feature and graph filters, it indicates that the performance of graph filters greatly depends on label spectra. Particularly, when the frequency response of a graph filter does not fit the label frequency, it might be inferior to all-pass filters, such as MLP. On the other hand, it poses *a principle of filter design: make feature response of filters be consistent with the main frequency band of label frequency as much as possible*. In terms of input information, it determines the performance of a filter - *if the frequency distribution of input vector is far from that of label vector, even an ideal filter would fail*. This observation is identical to our assumption in Section 3.4 - information of label vector is all we need and the distance between frequency distribution of input and label vectors reflects its usefulness. Therefore, $Er(g, \mathbf{y}_l)$ is the lower bound of $Er(g, \mathbf{x}_l)$ when $g(\tilde{A})$ are given. While an input vector may be useful for distinguishing one class, it may be helpless for another. In most GNNs, they tune the frequency distribution of features with a learnable linear transformation to generate a more informative input.

Here, we discuss the $Er(g, \mathbf{y}_l)$ of three types of filters:

$$Er(g_{l(\epsilon)}, \mathbf{y}_l)/R_l = \text{Var}(\mathbf{f}_l - \epsilon) + \mathbb{E}[\mathbf{f}_l - \epsilon]^2 = \text{Var}(\mathbf{f}_l) + (\mathbb{E}[\mathbf{f}_l] - \epsilon)^2 \qquad (6)$$

$$Er(g_{h(\epsilon)}, \mathbf{y}_l)/R_l = \text{Var}(2 - \mathbf{f}_l - \epsilon) + \mathbb{E}[2 - \mathbf{f}_l - \epsilon]^2 = \text{Var}(\mathbf{f}_l) + (\mathbb{E}[\mathbf{f}_l] + \epsilon - 2)^2 \qquad (7)$$

$$Er(g_{b(\epsilon')}, \mathbf{y}_l)/R_l \approx \frac{(\mathbb{E}[\mathbf{f}_l] + \epsilon' - 1)^4 + 6\text{Var}(\mathbf{f}_l)(\mathbb{E}[\mathbf{f}_l] + \epsilon' - 1)^2 + 8(1 - \epsilon')\text{Var}(\mathbf{f}_l)\mathbb{E}[\mathbf{f}_l]}{(1 + |\epsilon'|)^4}. \quad (8)$$

where we use $\text{Var}(\mathbf{f}_l^2) \approx 4\mathbb{E}[\mathbf{f}_l]^2\text{Var}(\mathbf{f}_l)$ derived from the delta method.

**Discussion.** An interesting observation is that *for a class with high dispersive spectrum, efforts of any single filters are to no avail*. From Corollary 3.3, we know that $\mathbb{E}[\mathbf{f}_l] = 1 - \tilde{\pi}_l$ and $\text{Var}(\mathbf{f}_l) = \tilde{\pi}_l^2 - (\tilde{\pi}_l)^2$. It demonstrates that higher homophily means lower $\mathbb{E}[\mathbf{f}_l]$, lower $\text{Var}(\mathbf{f}_l)$, and also lower prediction error for low-pass filters. On the other hand, we indicate that, in most cases, band-pass filters are more powerful than low-pass filters, let alone high-pass filters. However, *the prediction capacity of a signal filter is very limited when the means of spectra vary widely*.

## 4.2 HOW STRUCTURE MATTERS

Above, we catch a glimpse of spectral explanation of the behavior of graph filters. Below, we expand more understanding of graph filters. Assume that with learnable transformation, GNNs enable to generate an informative input. Here we discuss the prediction error of different graph filters under the optimal input $Y$. We revisit $Er(g, \mathbf{y}_l)$ using symmetric interaction matrix and propose a lower bound $er(g, \mathbf{y}_l)$ leveraging Proposition 3.1:

$$Er(g, \mathbf{y}_l) = \mathbf{y}_l^\top (I - g(\tilde{A}))^2 \mathbf{y}_l = R_l(I - 2\tilde{g}(\tilde{\Pi}) + \tilde{g}^2(\tilde{\Pi}))_{ll} \geq er(g, \mathbf{y}_l) = R_l(I - \tilde{g}(\tilde{\Pi})_{ll})^2. \quad (9)$$

**How structural information matters?** We indicate that, in the spatial point of view, graph filters can be interpreted as weight-tuning mechanisms on edges. The lower bound clearly demonstrates that a graph filter would have unsatisfactory prediction accuracy if it fails to make the homophily degree of the tuned graph large enough ($g[\tilde{\Pi}]_{ll}$ are far from 1).

Applying the prediction error lower bound to aforementioned specified filters, we have:

$$er(g_{l(\epsilon)}, \mathbf{y}_l) = (1 - \tilde{\pi}_l - \epsilon)^2 R_l; \quad er(g_{h(\epsilon)}, \mathbf{y}_l) = (1 + \tilde{\pi}_l - \epsilon)^2 R_l \quad (10)$$

$$er(g_{b(\epsilon')}, \mathbf{y}_l) = (1 + |\epsilon'|)^{-4} R_l(\epsilon'^2 - 2\epsilon'\tilde{\pi}_l + \tilde{\pi}_l^2)^2 \geq (1 + |\epsilon'|)^{-4} R_l(\epsilon' - \tilde{\pi}_l)^4. \quad (11)$$

**Discussion.** These error bounds indicate that: 1. a low-pass filter would fail on classes with low homophily degree - in turn, it confirms that the importance of homophily assumption for low-pass filters like GCN - it is identical with our spectral point of view; 2. high-pass filters have poor performances particularly on the high homophily graphs; 3. for a graph whose classes have consistent homophily degree (their self-interaction probabilities concentrate around a constant $\bar{\epsilon}$), $g_{b(\bar{\epsilon})}$ would work better than others. However, it is predictable that *any single filters would fail on graphs with diverse self-interaction probabilities*.

## 5 MODEL AND EMPIRICAL STUDY

Our theoretical analysis of graph information demonstrates that: 1. when node classes have inconsistent homophily degree or their label frequency distribution are far from each other, a single graph filter is prone to fail; 2. in most cases, band-pass filters would perform better than low-pass and high-pass filters; 3. a feature may contribute to the classification of one class but hinder the discrimination of another. Inspired by these, we propose a disentangled multi band-pass filter framework (DEMUF) which can be applied to any type of graphs no matter what kinds of graph information they have. The key point of our model is to learn multi band-pass filters which are used to capture different disentangled feature information respectively.

### 5.1 ARCHITECTURE OF TWO FRAMEWORKS OF DEMUF

Our framework includes *feature disentanglement* and *frequency filtering*. As we have emphasized the limitations of single filters, it is natural to leverage multi graph filters. Theoretically, piling up sufficient numbers of graph filters to capture all the frequency components can improve prediction performance. However, it is very expensive. To avoid this problem, we consider feature disentanglement - essentially, it is to disentangle frequency distributions of features into different families. Features in the same family are expected to have similar spectral properties, that is, they have similar frequency distributions or have overlap on their main frequency bands. Then for each family, we apply

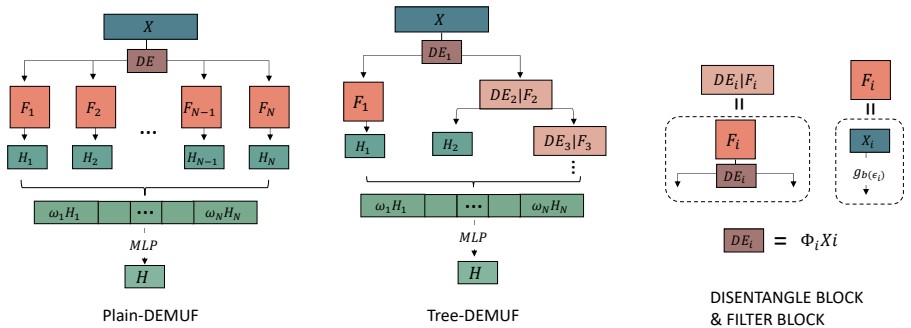

Figure 1: Illustration of Plain-DEMUF and Tree-DEMUF. There are two main model blocks of DEMUF frameworks: *disentangle block* and *filter block*. In Plain-DEMUF, all filter blocks run in parallel as their disentangled input are generated through a single disentangle block at the same time. Differently, each Tree-DEMUF layer contains two branches - one is early stopped while the other will be disentangled into two branches of the next layer after going through a filter.

a band-pass graph filter to capture their main frequency components. We propose two frameworks with different structures of filters: Plain-DEMUF and Tree-DEMUF (depicted in Fig. 1).

The DISENTANGLE block and FILTER block are formulated as follows:

$$X_k = \text{DISENTANGLE}(X, \Phi_k) = \Phi_k(X), \quad H_k = \text{FILTER}\Big(X_k, \epsilon_k, h_k\Big) = (g_{b(\epsilon_k)})^{h_k} X_k. \quad (12)$$

In our implementation, we provide two samples of DISENTANGLE functions $\Phi_k$: one is linear transformations, the other is GUMBEL_SOFTMAX (Jang et al., 2017) used to generate learnable masks for feature selection. In terms of the FILTER block, we use the band-pass filter defined in Definition 4.2, i.e., $g_{b(\epsilon)} = I - (1 + |\epsilon|)^{-2}(\tilde{A} - \epsilon I)^2$ as the identical filter form. Here, $\epsilon$ is the parameter of filter constrained in $[-1, 1]$ noting that $1 - \epsilon$ is the center of frequency response $g_{b(\epsilon)}$. In each FILTER block, $h$ is the number of layers. The framework of Plain-DEMUF with $N$ graph filters is:

$$H = \text{MLP}\Big(\text{CONCAT}\Big(\Big\{\text{FILTER}\Big(\text{DISENTANGLE}\Big(X, \Phi_k\Big), \epsilon_k, h_k\Big), \omega_k \Big| k = 1, \ldots, N\Big\}\Big)\Big).$$

Based on this, we implement a simple model called P-DEMUF. Precisely, we leverage a GUM-BEL_SOFTMAX to generate $N$ learnable masks $\{M_1, \ldots, M_N\}$ for feature sampling at once followed by different MLP. That is, $\Phi_k(X) = \text{MLP}_k(X \odot M_k)$.

Similarly, we develop a model, T-DEMUF, under the framework of Tree-DEMUF formulated by:

$$H_1, X_1 = \text{FILTER}\Big\{\Big(\text{DISENTANGLE}\Big(X, \Phi_1\Big), \epsilon, h\Big), \Big(\text{DISENTANGLE}\Big(X, \Psi_1\Big), \epsilon_1, h_1\Big)\Big\}$$

$$H_{k+1}, X_{k+1} = \Big\{\Big(\text{DISENTANGLE}\Big(X_k, \Phi_k\Big), \text{FILTER}\Big(\text{DISENTANGLE}\Big(X_k, \Psi_k\Big), \epsilon_k, h_k\Big)\Big\}$$

$$H = \text{MLP}\Big(\text{CONCAT}\Big(\Big\{\omega_k H_k, k = 1, \ldots, N\Big\}\Big)\Big).$$

In each T-DEMUF layer, we use GUMBEL_SOFTMAX with different parameters to generate two masks $M_k$ and $M'_k$ and $\Phi_k(X_k) = X_k \odot M_k$ and $\Psi_k(X_k) = X_k \odot M'_k$. In each layer, we stop further disentangling of the branch of $H_k$ by utilizing an additional constraint $\mathcal{L}(X_{k-1}, H_k) = \| X_{k-1} \odot M'_k - H_k \|_2^2$. Noting that $H_k = (g_{b(\epsilon_k)})^{h_k} X_{k-1} \odot M'_k$, this constraint is to make the main frequency bands of $H_k$ be consistent with frequency response of $(g_{b(\epsilon_k)})^{h_k}$.

**Model discussion.** Compared with filter-bank learning methods which directly apply an array of filters to features, our models use subsets of features. It can greatly reduce the amount of computation and parameters and help learning filters more efficiently and effectively. In addition, T-DEMUF uses an additional constraint to guide the filter learning process while P-DEMUF is a combination of multi graph neural networks which would not interfere with each other. Therefore, P-DEMUF is likely to obtain similar filters and require more filters to improve performance than T-DEMUF. The model visualization results in Fig. 2 validate this statement.

## 5.2 Experiments

To validate DEMUF, we compare the performances of P-DEMUF and T-DEMUF with that of spectral GNNs, spatial GNNs and MLP on extensive datasets.

### 5.2.1 Experiment Settings

**Datasets.** We use four types of real datasets - *Citation network*, *WebKB*, *Actor co-occurrence network* and *Wikipedia network*, to validate our proposed models. Cora and Citeseer (Sen et al., 2008) are widely used citation benchmarks which represent paper as nodes and citation between two papers as edges. Cornell, Texas, and Wisconsin (Pei et al., 2020) are three subgraphs of WebKB which is a webpage network with web pages as nodes and hyperlinks between them as edges. Chameleon and Squirrel (Rozemberczki et al., 2021) are two Wikipedia networks with web pages as nodes and links between pages as edges. The nodes originally have five classes while Bo et al. (2021) proposed a new classification criteria which divides nodes into three main categories. In this paper, the relabeled networks are called Chameleon2 and Squirrel2. Actor (Tang et al., 2009) is a subgraph of the fillm-director-actor-writer network whose nodes only represent actors and edges represent their collaborations. For all data, we use $60\%$ nodes for training, $20\%$ for validation and $20\%$ for testing. To intuitively show the homophily degree of a dataset, we calculate the mean of self-interaction probability (diagonal of interaction probability matrix) and show it in Table 1. This metric is similar to the node homophily in (Pei et al., 2020). More statistics of datasets can be found in Appendix A.

**Baselines.** We compare our models with four spectral GNNs: GCN (Kipf & Welling, 2017), ChebNet (Defferrard et al., 2016), GIN (Xu et al., 2019) (despite a spatial GNN, we can easily get its spectral form), ARMA (Bianchi et al., 2021). We list their spectral filter forms in Appendix A. In short, GCN is a well-known low-pass filter. The filter shape of GIN depends on its parameter $\epsilon$. In this paper, we fix $\epsilon = 0.3$ and thus it is also a low-pass filter. ChebNet and ARMA are high-order polynomial filters. In addition, we also add three spatial GNNs (whose spectral forms are hardly analyzed): GAT (Veličković et al., 2018), FAGCN (Bo et al., 2021), Geom_GCN (Pei et al., 2020). Both GAT and FAGCN utilize attention mechanism and FAGCN takes high frequency information into account. Geom_GCN is a novel aggregation method based on the geometry of graph (it is related because it was also empirically studied on graphs with different levels of homophily (Pei et al., 2020)). Finally, we also compare with MLP, a baseline without using any graph information.

**Experimental Setup.** For all experiments, we report the mean prediction accuracy on the testing data for 10 runs. We search learning rate, hidden unit, weight decay and dropout for all models in the same search space. Finally, we choose learning rate of 0.01, dropout rate of 0.5, and hidden unit of 32 over all datasets. The number of filters are searched between 2 to 10, and the final setting is: for T-DEMUF, we use 4 filters with 7 layers for Citation networks, 2 filters with 15 layers for all WebKB and Wikipedia networks, 5 filters with 1 layer for Actor. The numbers of MLP layers are 2, 2, 3 and 4, respectively. P-DEMUF uses: 3 filters with 8 layers for Citation networks; 5 filters for Cornell, 4 filters for Wisconsin and 3 filters for Texas - all of them are 1 layer; 7 filters with 9 layers for WebKB; 5 filters with 2 layers for Actor. P-DEMUF applies 2-layer MLP to all benchmarks. In addition, as the setting of benchmarks are the same as that in Geom_GCN, we refer to the results reported in Pei et al. (2020).

## 5.3 Result and Analysis

The experimental results are summarized in Table 1. Our models consistently outperform baselines over most benchmarks with significant improvement. On Cora and Citeseer, the datasets with a high level of homophily, our models are only comparable to GCN and other baselines. However, on all other datasets with a lower level of homophily, our models both obtained great performance gain.

To understand the impact of graph homophily on different types of graph filters, let us analyze the performance of all spectral GNNs. On high-homophily datasets, all GNNs perform similarly and the accuracy is much higher than MLP. That means the graph structure information is extremely useful in this case. However, on low-homophily datasets, many of them are even worse than MLP. GCN and GIN, the two low-pass filter based models, perform worst. The two GNNs with high-order graph filters, ChebNet and ARMA, are clearly superior to other models due to their higher spectrum coverage. However, they cannot beat our models with specially designed multiple filters. The

Table 1: **Node classification accuracy.** The first row is the mean of self-interaction probability.

| | | Cora | Cite. | Cornell | Texas | Wisc. | Cham. | Squi. | Cham.2 | Squi.2 | Actor |
|---|---|---|---|---|---|---|---|---|---|---|---|
| | #mean-$\pi_i$ | 0.861 | 0.809 | 0.436 | 0.356 | 0.413 | 0.338 | 0.290 | 0.516 | 0.425 | 0.393 |
| Spectral | GCN | 88.5 | 76.2 | 54.05 | 57.84 | 51.37 | 41.23 | 27.95 | 66.93 | 57.12 | 28.05 |
| | Cheb. | 88.21 | 76.26 | 80.00 | 78.38 | 78.43 | 51.71 | 36.52 | 75.44 | 66.11 | 35.76 |
| | GIN | 87.06 | 74.1 | 55.68 | 52.97 | 49.02 | 36.58 | 23.73 | 44.74 | 52.22 | 26.30 |
| | ARMA | 87.56 | 74.86 | 71.35 | 75.68 | 75.29 | 52.54 | 36.56 | 76.14 | 66.78 | 35.27 |
| Spatial | GAT | 88.32 | 76.85 | 55.14 | 61.08 | 54.51 | 47.46 | 32.66 | 70.92 | 61.42 | 29.32 |
| | FAGCN | **89.19** | 77.15 | 73.51 | 65.41 | 76.86 | 49.82 | 33.68 | 74.47 | 65.86 | 34.61 |
| | Geom_GCN | 85.27 | **77.9** | 60.81 | 67.57 | 64.12 | 60.90 | 38.14 | 73.20 | 63.30 | 31.63 |
| | MLP | 75.33 | 71.4 | 80.00 | 80.00 | 84.31 | 49.56 | 34.89 | 77.28 | 63.19 | 36.38 |
| Ours | T-DEMUF | 86.72 | 74.57 | 86.15 | 87.83 | 85.31 | **69.52** | **56.47** | **81.89** | **70.66** | 37.53 |
| | P-DEMUF | 87.85 | 75.69 | **86.49** | **89.73** | **89.68** | 69.46 | 56.45 | 79.17 | 68.87 | **37.68** |
| | | ↓1.34 | ↓2.21 | ↑6.49 | ↑9.73 | ↑4.37 | ↑8.62 | ↑18.33 | ↑4.61 | ↑3.88 | ↑1.30 |

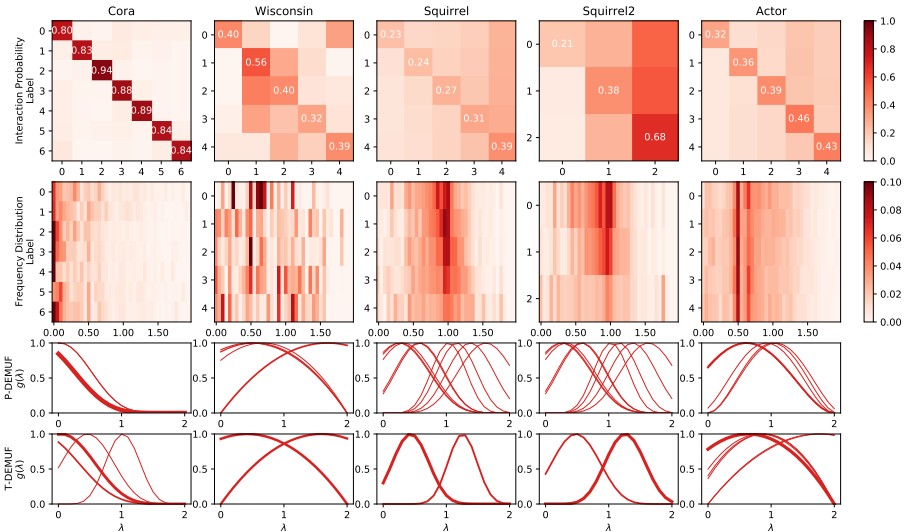

Figure 2: Visualizations of interaction probability matrix and frequency distribution of five datasets here and filters $g_{b(1-\epsilon)}^h$ learnt by P-DEMUF and T-DEMUF, where the thickness of curve reflects its weight $\omega$. It shows that frequency responses of our filters are consistent with labels frequency.

reason might be that the high complexity of their filters makes it more difficult to learn one optimal single filter. Finally, our model T-DEMUF yields over $18\%$ higher accuracy than the best baselines (Geom_GCN) on Squirrel; and P-DEMUF yields almost $10\%$ higher accuracy than MLP on Texas

In addition, we select some typical datasets and show the frequency distribution on these graphs in Fig. 2. We can obviously see that on Cora the spectrum is focused on low frequency components. This can explain why the low-pass filter based models can also perform well on it. On other datasets, the frequency distribution is more diverse, so the low-pass filters can not match with the important frequency components anymore. In contrast, both of our models, T-DEMUF and P-DEMUF, learn graph filters corresponding well to those components (as shown in the last two rows of Fig. 2). T-DEMUF uses fewer number of (more dispersed) filters but achieves comparable or better performance.

# 6    CONCLUSION

In this paper, we propose a theoretical analysis of graph information with the introduction of interaction probability and frequency distribution. We develop a deep understanding of how different structures and input influence the performance of graph filters. We also design a simple framework to learn a filter bank. Empirical results on extensive datasets validate the power of our model.

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

## A  BENCHMARKS AND MODEL DISCUSSION

### A.1  STATISTICS INFORMATION OF BENCHMARKS.

We provide statistics information of our benchmarks in Table. A.1.

Table 2: Datasets statistics.

| Dataset | Cora | Cite. | Cornell | Texas | Wisc. | Cham. | Squi. | Cham.2 | Squi.2 | Actor |
|---|---|---|---|---|---|---|---|---|---|---|
| # Nodes | 2708 | 3327 | 183 | 183 | 251 | 2277 | 5201 | 2277 | 5201 | 7600 |
| # Edges | 5429 | 4732 | 295 | 309 | 499 | 36101 | 217073 | 36101 | 217073 | 33544 |
| # Features | 1433 | 3703 | 1703 | 1703 | 1703 | 2325 | 2089 | 2325 | 2089 | 931 |
| # Classes | 7 | 6 | 5 | 5 | 5 | 5 | 5 | 3 | 3 | 5 |

### A.2  SPECTRAL FILTERS.

In our paper, we use four spectral GNN as baselines whose spectral filters are listed as Table.A.2 and define a band-pass filter $g_{b(\epsilon)}$ as a quadratic function with respect to the adjacency matrix.

Table 3: Spectral filters.

| Model | Filter |
|---|---|
| GCN | $\tilde{A}$ |
| GIN | $A + (I + \epsilon)I$ |
| ChebNet | $C^{(s)} = 2C^{(2)}C^{(s-1)} - C^{(s-2)};$ $C^{(2)} = 2L/\lambda_{max} - I; C_1 = I$ |
| ARMA | $(I + \sum_{k=1}^{K} q^k L^k)^{-1}(\sum_{k=0}^{K-1} p_k L^k)$ |
| Ours | $I - (1 + |\epsilon|)^{-2}(\tilde{A} - \epsilon I)^2$ |

Since $g_{b(\epsilon)}(\tilde{A}) = I - (1+|\epsilon'|)^{-2}(\epsilon I - \tilde{A})^2 = (1+|\epsilon|)^{-2}((1+|\epsilon|-\epsilon)I + A)((1+|\epsilon|+\epsilon)I - A)$, it is exactly an overlap between a low-pass filter $(1+|\epsilon|-\epsilon)I + A$ and a high-pass filter $((1+|\epsilon|+\epsilon)I - A)$. That is why $g_b(\epsilon)$ is a band-pass filter.

### A.3  MODEL DISCUSSION.

#### A.3.1  MOTIVATION OF DISENTANGLEMENT.

**Overlap if not disentangled.** Without disentanglement, it is highly possible that the learned filters have large overlaps if we do not induce any constraints on them. In our algorithm, we aim to train filters to capture the main frequency information of their input and assign different weights to that captured information depending on how much they contribute to label prediction. Therefore, it is natural to assume that if the input of filters is different, filters are less likely to overlap. Disentanglement can make the "input" different and more adaptable to each filter.

**Disentanglement reduces the model complexity.** With disentanglement, we divide node features into several subsets by learnable masking or map them into several low-dimensional spaces through linear transformations which lowers the dimension of corresponding features for each filter and meanwhile makes the input feature fits better to each filter.

### A.3.2 Motivation of T-DEMUF.

As we clarified in Section 5.1, our implementation of disentanglement is not random masking but learnable masking leveraging GUMBEL-SOFTMAX. These learnable maskings disentangle node features into several subsets of features. With a constraint $\mathcal{L}(X_{k-1}, H_k) =\| X_{k-1} \odot M'_k - H_k \|_2^2$, for each subset of features we train a band-pass filter to through their main frequency. As shown in Figure.2, this constraint guides the process of filter learning which can help reduce the overlap of filters' frequency responses such that can reduce the number of graph filters. At the same time, to minimize the supervised loss, maskings are trained to disentangle features whose frequency distribution are similar to those of labels and assign a higher weight; those useless captured feature information will be assigned a lower weight. Weights are also learnable.

### A.3.3 How can our filter bank selection be data-driven.

In our algorithm, although the form of our band-pass filter $g_b(\epsilon)$ is predefined, its parameters including $\epsilon$ and weight $\omega$ are learned from specified graphs. Moreover, the parameters of our feature disentanglement blocks (the linear transformations and learnable masking) are also learned from data which will affect the learning of the filter bank. Therefore, our filter bank selection is data-driven.

## A.4 More Experimental Results.

### A.4.1 Additional baseline - GPRGNN.

Here we compare our models with a related baseline GPRGNN(Chien et al., 2021). It is worth noting that the splitting in our paper is different from that in GPRGNN. In GPRGNN, its training set consists of the same number of nodes from each class while we just randomly choose our training data. We find that WebKB datasets are sensitive to the way of splitting due to their uneven distribution of labels ( the numbers of classes are: Cornell and Texas: 33/1/18/101/30, Wisconsin: 10/70/118/32/21). Although GPRGNN's splitting is likely better for model training, our model still outperforms it in the Wikipedia datasets, i.e., Chameleon and Squirrel. In GPRGNN's setting, the performances of MLP on WebKB are comparable to GPRGNNs' while in our setting, our proposed models' performances are much better than MLPs'. We also test T-DEMUF on Actor, Cornell, and Texas following the splitting of GPRGNN. As shown in Table.A.4.1, in most of the benchmarks, our model performs better than GPRGNN.

Table 4: Node classification accuracy of P-DEMUF and GPRGNN.

| Dataset | Different Splitting | | GPRGNN Splitting | | |
|---|---|---|---|---|---|
| | Cham. | Squi. | Cornell | texas | actor |
| GPRGNN | 69.52 | 49.93 | 91.36 | 92.92 | 39.30 |
| T-DEMUF | 67.48 | 56.47 | 91.26 | 92.79 | 41.11 |
| | ↑2.04 | ↑ 6.54 | ↓0.1 | ↓0.13 | ↑1.81 |

### A.4.2 Ablation study.

To show the advantage of using disentanglement, we provide an ablation study on five benchmarks. Here, we propose two ablation models based on P-DEMUF. Recall that the disentanglement block of P-DEMUF consists of learnable masking and linear transformations, we design our ablation models by taking off the component of masking and linear transformation. Also, for fair and intuitive comparison, we simply fix the number of filters as 2. The results shown as Table.A.4.2 validate that if we take off the disentanglement blocks of P-DEMUF, the results become worse in most of benchmarks.

Table 5: Node classification accuracy of P-DEMUF and its ablation models. We fix the number of band-pass filters as 2.

| Dataset | Cornell | Texas | Cham. | Squi. | Actor |
|---|---|---|---|---|---|
| P-DEMUF | 81.08 | 85.14 | 68.46 | 55.45 | 36.95 |
| P-DEMUF (w/o masking ) | 80.00 | 84.86 | 67.20 | 53.54 | 37.20 |
| P-DEMUF (w/o masking & linear) | 77.30 | 86.49 | 62.5 | 44.19 | 36.71 |

## B PROOF OF PROPOSITION

Here, we provide the proof of Proposition 3.2.

*Proof.* Since $\mathbf{x} = \sum_{i=0}^{n-1} \alpha_i \mathbf{u}_i$, $\mathbf{u}_i$ is the $i$-th unit eigenvector of $\tilde{L}$ and $\lambda^n = \mathbf{u}_i^\top \tilde{L}^n \mathbf{u}_i$ then we have

$$\mathbb{E}[\mathbf{f}^n] = \sum_{i=0}^{n-1} P(\mathbf{f} = \lambda_i)\lambda_i^n = \frac{\sum (\alpha_i \mathbf{u}_i)^\top \tilde{L}^n (\alpha_i \mathbf{u}_i)}{\sum \alpha_i^2} = \frac{\mathbf{x}^\top \tilde{L}^n \mathbf{x}}{\mathbf{x}^\top \mathbf{x}} = \frac{\mathbf{x}^\top (I - \tilde{A})^n \mathbf{x}}{\mathbf{x}^\top \mathbf{x}}. \quad (13)$$

$\square$

Below is the proof of Proposition 3.1.

*Proof.* For $P = D^{-1}A$ and $\tilde{A} = D^{-\frac{1}{2}}AD^{-\frac{1}{2}}$, and $\Pi$, $\tilde{\Pi}$ defined by Definition 3.1, the inequality can be represented as $(R\Pi^k + (\Pi^k)^\top R)_{lm} \geq 2(R^{\frac{1}{2}}\tilde{\Pi}^k R^{\frac{1}{2}})_{lm}$ which is equivalent to prove $\mathbf{y}_m^\top (P^k + (P^k)^\top)\mathbf{y}_l \geq 2\mathbf{y}_m^\top \tilde{A}^k \mathbf{y}_l$. Noting that, $(P^k)^\top = DP^k D^{-1}$ and $\tilde{A}^k = D^{\frac{1}{2}}P^k D^{-\frac{1}{2}}$, with $B = P^k + (P^k)^\top$, we have $B_{ij} = P_{ij}^k + \frac{d_i}{d_j}P_{ij}^k \geq 2\sqrt{\frac{d_i}{d_j}}P_{ij}^k = 2\tilde{A}_{ij}^k$. Therefore, $\mathbf{y}_m^\top (B - 2\tilde{A}^k)\mathbf{y}_l \geq 0$. Let $m = l$, then we get $\pi_l^k \geq \tilde{\pi}_l^k$.

To prove proposition b, we utilize lemma B.1. Since $g(\tilde{A})$ is symmetric, then we have

$$(g^2[\tilde{\Pi}])_{ll} = \frac{\mathbf{y}^\top (g(\tilde{A}))^2 \mathbf{y}}{\mathbf{y}^\top \mathbf{y}} \geq \left(\frac{\mathbf{y}^\top g(\tilde{A})\mathbf{y}}{\mathbf{y}^\top \mathbf{y}}\right)^2 = (g[\tilde{\Pi}]_{ll})^2.$$

$\square$

**Lemma B.1.** Let $B \in \mathbb{R}^{n \times n}$ is a symmetric matrix, $\forall ij$, $\mathbf{y} \in \mathbb{R}^n$, we have

$$\frac{\mathbf{y}^\top B^2 \mathbf{y}}{\mathbf{y}^\top \mathbf{y}} \geq \left(\frac{\mathbf{y}^\top B \mathbf{y}}{\mathbf{y}^\top \mathbf{y}}\right)^2.$$

*Proof.* Since $B$ is symmetric, then we have $B = U\Lambda U^\top$, here $U$ is matrix of unit eigenvectors of $B$. From the proof of Proposition 3.2, we obtain that $\frac{\mathbf{y}^\top B^2 \mathbf{y}}{\mathbf{y}^\top \mathbf{y}} = \frac{\sum (\alpha_i \lambda_i)^2}{\sum \alpha_i^2}$ and $\left(\frac{\mathbf{y}^\top B \mathbf{y}}{\mathbf{y}^\top \mathbf{y}}\right)^2 = \frac{(\sum \alpha_i^2 \lambda_i)^2}{(\sum \alpha_i^2)^2}$. From Hölder's inequality, we have $(\sum (\alpha_i \lambda_i)^2)(\sum \alpha_i^2) \geq (\sum \alpha_i^2 \lambda_i)^2$. Therefore, we have $\frac{\sum (\alpha_i \lambda_i)^2}{\sum \alpha_i^2} \geq \frac{(\sum \alpha_i^2 \lambda_i)^2}{(\sum \alpha_i^2)^2}$. $\square$

