# OpenReview forum: "Graph Information Matters: Understanding Graph Filters from Interaction Probability"
_ICLR.cc/2022/Conference — ICLR 2022 Submitted_

### Official Review · Reviewer_kKsu · 2021-11-01

**Correctness:** 2
**Technical Novelty And Significance:** 2
**Empirical Novelty And Significance:** 2
**Recommendation:** 3
**Confidence:** 4

**Main Review:**

This paper suffers on many fronts.
I wish I could give a more thorough review of this work, but the writing is so poor that assessing the merits of this work would be quite difficult.
The authors make many statements with very little explanation.
For instance, Proposition 3.1 is largely a jumble of symbols with very little surrounding discussion: it is not explained why I should remotely care about this result.
Section 3.4 states things that are largely obvious from a graph signal processing perspective.
Definition 4.1 is a strange way to look at prediction error: why are you using a Frobenius norm to evaluate the error of a node classificiation problem?
The conclusions reached in Section 4.1 are quite obvious and well-understood in the broad literature.
Of course filters should "match the true signal," this is the most basic fact from signal processing!

The conclusion that graph neural networks should use a filter bank rather than a single filter is nothing new.
See, for instance, the work "Stability Properties of Graph Neural Networks" from Gama et. al. (2020).

Overall, the contributions of this paper are extremely lacking.
And even if there was a notable contribution of this work, the writing quality obscures it so much for it to be at all useful.
I would also like to point out that the authors failed to cite even the most widely-known introductory papers for graph signal processing, such as that of Ortega et. al. (2018).
The failure of the authors to do so indicates a severe lack of familiarity with the graph signal processing literature, which this paper claims to contribute to.

**Summary Of The Paper:**

This paper studies the applications of graph filters in graph neural networks, and attempts to understand the links between graph spectral analysis, random walks, and homophily.
The authors introduce "interaction probability" as a metric for understanding homophily and its relationship to random walk transition matrices.
In this framework, the authors claim to understand a notion of graph information, as well as an understanding of how the performance of graph filters depends on both graph structure and input signals, with applications to graph neural networks.

**Summary Of The Review:**

This paper does not live up to its claims, and is extremely poorly written.
The authors spent most of their effort on this work stating obvious things in convoluted ways, amounting to very limited statements regarding graph filters.
Despite their claims of reaching "deep understanding" of graph filters, this paper achieves very little.
Due to the poor writing and shallow application of this work, I strongly recommend this paper be rejected.

---

> ### Author Response · Authors · 2021-11-23
> **Response to Reviewer kKsu**
>
> Thank you for the constructive comments, but we kindly disagree with them with the following evidence.
>
> **Q1 Writing quality**
>
> The other two reviewers’ opinions clearly contradict this. Reviewer RH1y: “Overall, the introduction part is well written, and the studied problem is novel and interesting.”
> Reviewer nP66: “The paper is well-written and clear to understand.”
>
> **Q2 Proposition 3.1**
>
> In our revision, we add more discussion about proposition 3.1 and interaction probability.
> In short, in Proposition 3.1 we propose two inequalities - one is to show the symmetric self-interaction probability is the lower bound of the original self-interaction probability; another is to illustrate a useful inequality of symmetric self-interaction probability which is leveraged to derive a lower bound of our prediction error in Section 4.2. $g(\Pi)$ is a function with respect to the interaction probability matrix $\Pi$ while $g[\Pi]=R^{-1}Y^\top g( P)Y$, that is, $g$ is not applied to $\Pi$ but applied to the transition probability matrix $P$. When $Y$ is a square matrix, $g[\Pi]=g(\Pi)$, otherwise they are unequal. However, concerning that the notation of $g[\Pi]$ sounds confusing, we revise the notation as $\tilde{g}(\Pi)$ in our revision. The connection between Proposition 3.1 and the proposed algorithms is: Proposition 3.1 is directly used for our following theoretical analysis which inspires the proposed algorithms.
>
> **Q3 “The conclusions reached in Section 4.1 are quite obvious and well-understood in the broad literature. Of course, filters should "match the true signal," this is the most basic fact from signal processing!”**
>
> We all know the filter “should match the true signal”, but we should also understand better why and how to match? In this paper, we show that the performance of a graph filter (in the context of GNNs) should be related to the interaction probability, i.e. the graph structure as well as node labels. So, in fact, we are not directly matching the “signal”, but just designing learnable filters to match the graph structure and labels. The filters are then learned from the data and automatically matched with the input features, i.e. the true signals.
>
>
> **Q4 The conclusion that graph neural networks should use a filter bank rather than a single filter is nothing new. See, for instance, the work "Stability Properties of Graph Neural Networks" from Gama et. al. (2020).**
>
> We checked this reference, and it is clearly not relevant. That paper is about stability, i.e. the change over some perturbation; even its conclusion is that using a filter bank is better than a single filter, which is only for stability. Our paper instead focuses on analyzing the final performance of one filter w.r.t a given graph structure and labels. The conclusion is more than “a single filter is not enough”. We also provide a more detailed analysis about when a filter (or a special GNN, such as GCN) can perform well or badly. Using filter banks is also just one of the solutions, and in many cases a low-pass filter based GNN is enough.
>
>
>
> **Q5. Reference about “ Ortega et. al. (2018)”.**
> Thanks for pointing out the missing references. We appreciate the works in graph signal processing and add this reference in our revised version. However, we do not think it is fair enough to judge a paper only from a missing reference (especially when this reference is only a fundamental paper but not a directly related method). Graph signal processing is very related, but as we claimed in the paper, “In this paper, we focus on the analysis of graph filters in the context of graph neural networks.”

---

> > ### Comment · Reviewer_kKsu · 2021-11-29
> > **Your responses have slightly clarified some concerns**
> >
> > Reading some of the other responses, I see that the authors have clarified some concerns that came up. However, I still feel that the presentation of this work is not done very well, and is clouded by excessive notation. I also don't think that the authors' approach yields any particular novel insight regarding the role of graph filters in GNNs beyond the existing literature. I am willing to raise my score slightly, but still do not think this paper should be accepted to ICLR.

---

> > > ### Author Response · Authors · 2021-11-30
> > > **Response**
> > >
> > > Thank you for the reply and raising the score, but most of your concerns have been clarified in the rebuttal, and we feel that the two major reasons for your low scores do not have enough evidence.
> > >
> > > 1. Presentation: First, we do not understand why the concerns are clarified from "reading other responses", because the problems the reviewer raised have been directly answered under your reviews. Moreover, what we want to emphasize is the clear contradiction to your statement about "this paper is extremely poorly written". We admit that the notations are somehow complex (but still accurate), and we are willing to improve it as another reviewer nP66 suggested, however, it seems that the general story of the paper is well received by other reviewers.
> > >
> > > 2. Novelty: The reviewer said "The authors spent most of their effort on this work stating obvious things in convoluted ways". However, it is obvious that our claim about the relation between the graph filters and interaction probability is not an obvious thing. The evidence the reviewer listed is not supportive of this conclusion at all. Please refer to our answer to Q4 and Q3. As far as we know, there is no similar research on analyzing the performance of graph filters in the context of GNNs from the perspective of interaction probabilities.
> > >
> > > In sum, we appreciate the reviewer’s effort, but we kindly request the reviewer to give us a more fair evaluation regardless of his/her background.
> > >
> > > Many seemingly "obvious things" require evidence or theoretic analysis. Discovering the reasons for these obvious things is not trivial. Take a simple example, from the knowledge of graph signal processing, it seems obvious that GCNs are low-pass filters, and deep GCNs can lead to over-smoothing; but when researchers re-discovered the phenomenon and analyzed its underlying reasons, those works became impactful and lead to various new models.

---

### Official Review · Reviewer_nP66 · 2021-11-02

**Correctness:** 3
**Technical Novelty And Significance:** 3
**Empirical Novelty And Significance:** 1
**Recommendation:** 6
**Confidence:** 5

**Details Of Ethics Concerns:**

GNNs can be combined with social network graphs to predict information about users which otherwise, they might have kept private. Thereby, I would encourage the authors to put an Ethics Statement in their paper as per the ICLR guidelines (https://iclr.cc/Conferences/2022/AuthorGuide).

**Main Review:**

The paper is well-written and clear to understand.

## Strengths:
1. The theoretical analysis looks clean and simple. Equation 6, 7, 8, 9 and 10 provide a good indication of how the prediction error is dependent on pi_l. It also gives insight into the homophily requirements for good performance.
2. The proposed model is simple and gives good performance on datasets of varying homophily scores.

## Weaknesses:
1. The error analysis is done on MSE rather than Softmax (which I suspect is what is used in the experiments). However, this doesn't take away much from the presented work and the analysis still feels insightful enough.
2. The motivation for disentanglement block is unclear. The paper states *Theoretically, piling up sufficient numbers of graph filters to capture all the frequency components can improve prediction performance. However, it is very expensive. To avoid this problem, we consider feature disentanglement - essentially, it is to disentangle frequency distributions of features into different families. Features in the same family are expected to have similar spectral properties, that is, they have similar frequency distributions or have overlap on their main frequency bands.* While it is clear that it will be difficult to have large number of band-pass filters, it is unclear how disentanglement of features work to avoid this scenario.
3. Implementation of disentanglement seems to be just random masking. How does it satisfy the statement made *Features in the same family are expected to have similar spectral properties* ?
4. The motivation for P-DEMUF is pretty clear, but the T-DEMUF architecture motivation is not clear and there is no mention of it anywhere. Also, the equations of T-DEMUF are somewhat confusing. It feels like there might be a typo there, at least by looking at Figure 1.
5. GPRGNN by [Chien et. al] is missing from baseline comparison.

In light of the above, I am inclined to rate this paper marginally below threshold.

There are few typos in the paper:
1. Var(f_l) in Section 3.4 and in other places it appears has a mistake.
2. Equations for T-DEMUF likely have a typo in there.

**References:**

Eli Chien, Jianhao Peng, Pan Li and Olgica Milenkovic. Adaptive Universal Generalized PageRank Graph Neural Network. ICLR, 2021.

**Summary Of The Paper:**

It is observed that for Graph Neural Networks (GNNs), the correlation between the labels and the graph structure matters. The paper gives a theoretical analysis of this behavior via interaction probability and frequency distribution. The analysis shows why homophily is favorable for GNNs. Additionally, they are also able to identify some conditions when GNNs can do well. But the key point made is that it is not possible to cater to all possible scenarios with a single filter. The paper further goes to propose a model that builds multiple band-pass filters over the range of [-1, 1] and then aggregates information over these filters to give improved performance.

**Summary Of The Review:**

The paper is well-written and clear to understand. Theoretical analysis is clean, simple and gives interesting insights. The results are also very strong with good improvements on several datasets. However, there are several weaknesses in the paper A] the motivation for the disentanglement block is unclear B] it is also unclear how simple random masking achieves disentaglement C] The motivation for T-DEMUF is unclear as well and D] a very related baseline by [Chien et. al] is missing from comparison.

In light of the above, I am inclined to rate this paper marginally below threshold.

**References:**

Eli Chien, Jianhao Peng, Pan Li and Olgica Milenkovic. Adaptive Universal Generalized PageRank Graph Neural Network. ICLR, 2021.

---

Update post response phase:
Having gone through other reviews, it seems that there maybe a lack of clarity in the paper, particularly on the theoretical aspects. I admit it was a bit of an effort to keep up with all the notations and the paper will definitely strengthen by bringing in lot more clarity in notations.

Having said that, I would like to add that although the conclusion of the work may not be new, but new perspectives for the same conclusion should not be discouraged. In mathematics, we have multiple proofs for a single statement like "There are infinite number of primes" [A], each of them brings a new perspective and has something to offer. I believe that we would be doing a disservice if we rejected papers simply on the basis that conclusion is not new, without really analyzing if the paper is offering a newer fresh perspective or not.

However, beyond these issues, I believe it is important to assess the actual model strengths as well. This is mostly where I find many things not so clear, for example, the feature disentanglement aspect is not clearly explained. Also, how does T-DEMUF work is not very well explained and what is the exact motivation for T-DEMUF is also quite lacking.

In light of these, I am inclined to retain my current assessment of marginally below acceptance threshold.

[A] https://primes.utm.edu/notes/proofs/infinite/

---

Update post second response phase:

The authors have addressed all my issues. However, the outcome is the following:
1. The notations have to be improved in the paper
2. Explanation and motivation for disentanglement has to be added to the paper
3. Explanation and motivation for T-DEMUF has to be added to the paper.

However, I believe these should not be difficult changes to make and hence I am willing to change my rating to marginally above acceptance threshold, under the assumption that the authors would make these changes in the final revision.

---

> ### Author Response · Authors · 2021-11-23
> **Response to Reviewer nP66 - Part I**
>
> Thank you for the reviewer’s comments and suggestions.
>
> **Q1. The error analysis is done on MSE rather than Softmax (which I suspect is what is used in the experiments). However, this doesn't take away much from the presented work and the analysis still feels insightful enough.**
>
> We agree with this point. However, it is common to take off some non-linear components to simplify model analysis in deep learning literature. Since MSE is also a commonly used error evaluation, as the reviewer said, although there is a gap between the theoretical analysis and practical implementation, it will not devalue our insight.
>
> **Q2. The motivation for the disentanglement block is unclear. While it is clear that it will be difficult to have a large number of band-pass filters, it is unclear how disentanglement of features works to avoid this scenario.**
>
> (1)Overlap if not disentangled. Without disentanglement, it is highly possible that the learned filters have large overlaps if we do not induce any constraints on them. In our algorithm, we aim to train filters to capture the main frequency information of their input and assign different weights to that captured information depending on how much they contribute to label prediction. Therefore, it is natural to assume that if the input of filters is different, filters are less likely to overlap. Disentanglement can make the “input” different and more adaptable to each filter.
>
>  (2) Disentanglement reduces the model complexity. It lowers the dimension of corresponding features for each filter and meanwhile makes the input feature fits better to each filter.
> In our revised paper, we added more explanation of the motivation, and we also added an ablation study to show the advantage of using disentanglement. The disentanglement block of P-DEMUF consists of learnable masking and linear transformation. We fix the number of filters as 2 and propose two ablation models by taking off the masking and linear transformation of P-DEMUF. The results are shown in Appendix. We also list them as follows for your reference:
>
> | Data | Cornell | Texas |Chameleon |Squirrel |Actor|
> |--|--|--|--|--|--|
> | P-DEMUF | 81.08 | 85.14 | 68.47 | 55.45 | 36.95 |
> | P-DEMUF (w/o masking ) | 80.0 | 84.86 | 67.20  | 53.54 | 37.20 |
> | P-DEMUF (w/o masking & linear) | 77.30 | 86.49 | 62.5 | 44.19 | 36.71 |
>
> Take the Cornell dataset as an example, if we take off the disentanglement blocks of P-DEMUF, the result becomes worse than either P-DEMUF.
>
>
> **Q3. The motivation for P-DEMUF is pretty clear, but the T-DEMUF architecture motivation is not clear and there is no mention of it anywhere. Implementation of disentanglement seems to be just random masking.  It is also unclear how simple random masking achieves disentanglement. How does it satisfy the statement made Features in the same family are expected to have similar spectral properties?**
>
> As we clarified in Section 5.1, our implementation of disentanglement is not random masking but learnable masking leveraging GUMBEL-SOFTMAX. These learnable maskings disentangle node features into several subsets of features. With a constraint $\mathcal{L}(X_{k-1},H_k) = \parallel X_{k-1}\odot M'_k-H_k\parallel_2^2$, for each subset of features we train a band-pass filter to through their main frequency. As shown in Figure.2, this constraint guides the process of filter learning which can help reduce the overlap of filters’ frequency responses such that can reduce the number of graph filters. At the same time, to minimize the supervised loss, maskings are trained to disentangle features whose frequency distribution are similar to those of labels and assign a higher weight; those useless captured feature information will be assigned a lower weight. Weights are also learnable.

---

> > ### Comment · Reviewer_nP66 · 2021-11-23
> > **Q2 and Q3**
> >
> > Thank you for the responses to the various concerns.
> >
> > > Overlap if not disentangled. ... Therefore, it is natural to assume that if the input of filters is different, filters are less likely to overlap. Disentanglement can make the “input” different and more adaptable to each filter.
> >
> > It is clear to me that, with disentanglement, one can get diverse set of filters. However, the paper seems to be implicitly claiming the following statement: "We do not need large number of band-pass filters, we only need to learn handful of diverse band-pass filters." It is not clear to me why this should be true? What I infer from your response is that disentanglement helps you identify those handful of diverse band-pass filters, rather than it helping to avoid the scenario of utilizing large number of band-pass filters. For me, this is the disconnect between the explanations and these parts become unclear.
> >
> > > Disentanglement reduces the model complexity. It lowers the dimension of corresponding features for each filter and meanwhile makes the input feature fits better to each filter.
> >
> > This is interesting.
> >
> > > In our revised paper, we added more explanation of the motivation, and we also added an ablation study to show the advantage of using disentanglement. The disentanglement block of P-DEMUF consists of learnable masking and linear transformation.
> >
> > Thanks, this is useful and it explains how useful the disentanglement block is.
> >
> > > our implementation of disentanglement is not random masking but learnable masking leveraging GUMBEL-SOFTMAX.
> >
> > Can you expand on this and give precise details about how GUMBEL-SOFTMAX is leveraged to learn masks? Are you doing GUMBEL-SOFTMAX sampling and using straight through differentiation for training or some such? If so, please clarify these details in the paper.
> >
> > Finally, I am still not sure I get the motivation for T-DEMUF. It was easy to understand how you might have come up with P-DEMUF architecture. But the T-DEMUF is very unusual and you may want to explain what was the motivation for that design and what were the expectations from such a Tree-like design.

---

> > > ### Author Response · Authors · 2021-11-24
> > > **Response to Q2 and Q3**
> > >
> > > > What I infer from your response is that disentanglement helps you identify that handful of diverse band-pass filters, rather than it helps to avoid the scenario of utilizing a large number of band-pass filters. For me, this is the disconnect between the explanations and these parts become unclear.
> > >
> > > 1. According to our theoretical analysis, given the frequency distributions of labels, we know which frequency bands are useful for prediction and we need to design filters such that their frequency response can cover most of the target frequency bands. If a handful of filters highly overlap, then we are much likely to use more filters than the handful of diverse filters. Although there is no strong guarantee that disentanglement can directly reduce the number of filters, practically, disentanglement can help us use fewer filters to achieve satisfactory performance,
> > >
> > > > Are you doing GUMBEL-SOFTMAX sampling and using straight through differentiation for training or some such?
> > >
> > > Yes, we use GUMBEL-SOFTMAX to do sampling and differentiation for training. The parameter is the log probabilities of GUMBEL-SOFTMAX. We will clarify it in the revision.
> > >
> > > > Finally, I am still not sure I get the motivation for T-DEMUF...you may want to explain what was the motivation for that design and what were the expectations from such a Tree-like design.
> > >
> > > There are two main steps of our models - train filters to capture the main frequency information of features and learn transformations to make the captured frequencies closed to the target frequencies - the main frequency of labels. According to our theoretical analysis (recall the discussion in Section 4.1 and 4.2), dispersive spectra are a challenge to filter learning. Therefore, instead of directly feeding all features into filters, features with similar frequency distribution are expected to be fed to the same filter and transformation. With disentanglement, P-DEMUF can learn diverse filters but it may divide “similar” features apart. Theoretically, the more disentanglement we have,  the better filters can learn. However, the improvement gain may be limited along with the high cost. Therefore, we need to introduce an additional constraint in T-DEMUF to identify “similar” features - for a collection of features if there exists a filter such that most of the frequency information can go through it then we say these features have common main frequencies. In T-DEUF, if a collection of features is identified to be similar, then we will stop disentanglement and filtering (the left branch), otherwise, we keep filtering and disentanglement.

---

> > > > ### Comment · Reviewer_nP66 · 2021-11-27
> > > > **Q2 and Q3**
> > > >
> > > > Thanks for the clarifications. It would be good to add the motivation for T-DEMUF in the paper itself. Also, the explanation given for disentanglement.

---

> ### Author Response · Authors · 2021-11-23
> **Response to Reviewer nP66 - Part II**
>
> **Q4. GPRGNN by [Chien et. al] is missing from baseline comparison.**
>
> The splitting in our paper is different from that in GPRGNN. In GPRGNN, its training set consists of the same number of nodes from each class while we just randomly choose our training data. We find that WebKB datasets are sensitive to the way of splitting due to their uneven distribution of labels ( the numbers of classes are: Cornell and Texas: 33/1/18/101/30, Wisconsin: 10/70/118/32/21). Although GPRGNN’s splitting is likely better for model training, our model still outperforms it in the Wiki dataset, i.e., Chameleon (our 69.52% v.s. GPRGNN 67.48%) and Squirrel (our 56.47% v.s. GPRGNN 49.93%). In GPRGNN’s setting, the performances of MLP on WebKB are comparable to GPRGNNs’ while in our setting, our proposed models' performances are much better than MLPs’.  We also test T-DEMUF on Actor, Cornell, and Texas following the splitting of GPRGNN. On Actor, our accuracy is 41.11% while GPRGNN’s is 39.30%; on Cornell, our accuracy is 91.26% and GPRGNN’s is 91.36%; on Texas, our accuracy is 92.79% while GPRGNN’s is 92.92%. Therefore, in most of the benchmarks, our model performs better than GPRGNN.
>
> **Q5.There are a few typos in the paper: 1. $\text{Var}(\mathbf{f}_l)$ in Section 3.4 and in other places, it appears to have a mistake. 2 . Equations for T-DEMUF likely have a typo in there.**
>
> 1. We check the calculation of $\text{Var}(\mathbf{f}_l)$ carefully without finding a mistake. $\text{Var}(\mathbf{f}_l)=\tilde{\pi}_l^2-(\tilde{\pi}_l)^2$, recall the remark of Def.3.1. and Proposition 3.1.b, $\tilde{\pi}_l^2\geq(\tilde{\pi}_l)^2$, therefore $\text{Var}(\mathbf{f}_l)$ is non-negative.
>
> 2. Also, there is no typo in the formulation of T-DEMUF. We suppose that the reviewer may be confused by the first layer of T-DEMUF which is $H_1,X_1 = \textrm{FILTER}\Big[\Big(\textrm{DISENTANGLE}\Big(X,\Phi_1\Big),\epsilon,h\Big),\Big(\textrm{DISENTANGLE}\Big(X,\Psi_1\Big),\epsilon_1,h_1\Big)\Big]$ which is equal to $H_1,X_1 = \textrm{FILTER}\Big(\textrm{DISENTANGLE}\Big(X,\Phi_1\Big),\epsilon,h\Big),\textrm{FILTER}\Big(\textrm{DISENTANGLE}\Big(X,\Psi_1\Big),\epsilon_1,h_1\Big)$. And it is consistent with what we discript in Figure 1 - for each right branch, it will go through a graph filter and then be disentangled into two sub-branches.

---

> > ### Comment · Reviewer_nP66 · 2021-11-23
> > **Q4 and Q5**
> >
> > > our model still outperforms it in the Wiki dataset, i.e., Chameleon (our 69.52% v.s. GPRGNN 67.48%) and Squirrel (our 56.47% v.s. GPRGNN 49.93%). In GPRGNN’s setting, the performances of MLP on WebKB are comparable to GPRGNNs’ while in our setting, our proposed models' performances are much better than MLPs’
> >
> > The numbers quoted here are switched up from what is reported in the Appendix, which is the correct version?
> >
> > > We check the calculation of $\textrm{Var}(\textbf{f}_l)$ carefully without finding a mistake. $\textrm{Var}(\textbf{f}_l) = \tilde{\pi}_l^2 - (\tilde{\pi}_l)^2$, recall the remark of Def.3.1. and Proposition 3.1.b, $ \tilde{\pi}_l^2 \geq (\tilde{\pi}_l)^2$, therefore $\textrm{Var}(\textbf{f}_l)$ is non-negative.
> >
> > I am a bit confused. Is there a difference between $\tilde{\pi}_l^2$  and $(\tilde{\pi}_l)^2$? I thought it was a bernoulli-like parameter and hence its variance should be of the form $p(1-p) = p - p^2$, so I suspected that $\textrm{Var}(\textbf{f}_l) = \tilde{\pi}_l - (\tilde{\pi}_l)^2$.
> >
> > > We suppose that the reviewer may be confused by the first layer of T-DEMUF which is
> >
> > No my confusion actually came from the definition of $H_{k+1}$ and $X_{k+1}$. The definition is given as,
> >
> > $H_{k+1}, X_{k+1} = \left\\\{ (\textrm{DISENTANGLE}(X_k, \Phi_k), \textrm{FILTER}(\textrm{DISENTANGLE}(X_k, \Psi_k), \epsilon_k, h_k) ) \right\\\}$
> >
> > I was under the impression that even $H_{k+1}$ should have a filter term, but I guess that is not the case.

---

> > > ### Author Response · Authors · 2021-11-24
> > > **Response to Q4 and Q5**
> > >
> > > Thank you for your instantly reply!
> > >
> > > > The numbers quoted here are switched up from what is reported in the Appendix, which is the correct version?
> > >
> > > The numbers in the rebuttal are correct. Actually, we provide the performance gain in the last row of Table A.4, which shows that our model yields 2.04% higher accuracy on Chameleon. So, it is a typo of the result of Chameleon we list in Table.A.4. Sorry for the typo and we will correct it in a future version.
> > >
> > > > Is there a difference between $\tilde \pi_l^2$ and $(\tilde \pi_l)^2$? I thought it was a Bernoulli-like parameter and hence its variance should be of the form $p(1-p) = p - p^2$.
> > >
> > > As we clarified in Remark.1 and demonstrated in Proposition 3.1.b, $\tilde \pi_l^2=\frac{\mathbf{y}_l^\top\tilde A^2\mathbf{y}_l}{\mathbf{y}_l^\top\mathbf{y}_l}$, the $2$-step symmetric interaction probability, is different from $(\tilde \pi_l)^2=(\frac {\mathbf{y}_l^\top\tilde A\mathbf{y}_l}{\mathbf{y}_l^\top\mathbf{y}_l})^2$ which is the square of $1$-step symmetric interaction probability $\tilde \pi_l$.
> > >
> > > > I was under the impression that even $H_\{k+1\}$ should have a filter term...
> > >
> > > In the algorithm of T-DEMUF, we want all the disentangled features to be filtered at least once, that is why we assign a filter on $H_1$. Since $H_\{k+1\}$ is a disentanglement of $X_k$ which is already filtered, there is no need to assign an additional filter term and we just early stop the branch of $H_\{k+1\}$.

---

> > > > ### Comment · Reviewer_nP66 · 2021-11-27
> > > > **Q4 and Q5**
> > > >
> > > > Thanks for the clarifications. Given the confusion with the notations, it might be worthwhile to have different notation that would not cause confusion.

---

### Official Review · Reviewer_RH1y · 2021-11-02

**Correctness:** 2
**Technical Novelty And Significance:** 3
**Empirical Novelty And Significance:** Not applicable
**Recommendation:** 5
**Confidence:** 4

**Details Of Ethics Concerns:**

None.

**Main Review:**

Here are the detailed comments:

[Problem] Very nice introduction and literature review. The problem is novel and interesting.

[Notation] The notation set shall be largely simplified. For example, y_i, c_m, r_i all refer to the node labels.

[Typos] There are typos throughout this paper. For example, missing space in the 9th line of Sec 3.3; in Sec 3.4, graph frequency -> graph frequencies? For {\alpha_i = <u_i, x>}_i, why does the index i appear in both the inside and the outside of the parentheses?

[Theoretical Analysis] In sec 3.3, the authors claimed that \phi_i^k = \sum P_{i,j}^k is the probability that a random walker starting from v_i and stays in C_l. Based on my understanding, \phi_i^k does not exclude the case that the random walk traverse outside of C_l. Same in the Def. of interaction probability. Does the interaction probability also consider the case that random walk traverse to some other communities (different from l and m)? If so, please provide the justifications - what is the motivation and rationale of these formulations.

[Proposition 3.1] There lack of insightful discussions about Proposition 3.1. What are the physical meanings of g(\Phi) and g[\Phi]? What does Proposition 3.1 tell us? And, what is the connection between Proposition 3.1 to the proposed algorithms?

[Confusiong Notions] In Def. 3.2, what do you mean about the distributional representation? The authors may want to provide further explanation. In Proposition 3.2, "f be the frequency of signal x" --> "f be the frequency distribution of signal x"?

[Over-calimed Contribution] (1) The low-pass/high-pass filters seem very similar to "Beyond Low-frequency Information in Graph Convolutional Networks". Moreover, the authors shall provide theoretical proofs to show why the g_b is a band-pass filter. (2) The authors claimed that this paper provides a "data-driven" mechanism for filter bank selection, which is not well described in the algorithm description. (3) the theoretical analysis and algorithm feel disconnected.

**Summary Of The Paper:**

The paper presents an insightful analysis of the induced graph filters in GNNs. To accommodate the heterogeneity of graphs, the authors provide a family of novel GNNs for learning data-specific filter banks. Overall, the introduction part is well written, and the studied problem is novel and interesting. However, the technical part of this paper has some issues: (1) the contributions of this paper are over-claimed, (2) the theoretical analysis and algorithm are disconnected, (3) mistakes in the theoretical results.

**Summary Of The Review:**

The studied problem of this paper is interesting, while there may exist some issues in the theoretical foundation of the proposed algorithm.

---

> ### Author Response · Authors · 2021-11-23
> **Response to Reviewer RH1y - Part I**
>
> **Q1. Notation and typos.**
>
> Thanks for your suggestion. In our revision, we simplified the notations and corrected those typos.
>
> **Q2. Theoretical analysis based on the interaction probability.**
>
> *In sec 3.3, the authors claimed that $\pi_i^k = \sum P_\{i,j\}^k$. Based on my understanding, $\pi_i^k$ does not exclude the case that the random walk traverse outside of $\mathcal{C}_l$. Same in the Def. of interaction probability. Does the interaction probability also consider the case that random walk traverse to some other communities (different from l and m)? If so, please provide the justifications - what is the motivation and rationale of these formulations.*
>
> In our paper, we denote $\pi^k_i(\mathcal{C}_l)$ as the probability that a random walker starting from $v_i$ stays in $\mathcal{C}_l$ at the $k$-th step and there is no need to exclude the case the random walker traverse other classes as long as the walker will arrive $\mathcal{C}_l$ at the $k$-th step. Therefore,  $\pi^k_i(\mathcal{C}_l)=\sum\limits_\{j\in\mathcal{C}_l\}P_\{ij\}^k$. Actually, $\pi^k_i(\mathcal{C}_l)$ is exactly the proportion of  $\mathcal{C}_l$
>  in the $k$-hop neighbors of node $v_i$.  Similarly, the interaction probability $\Pi_\{lm\}^k$ is the mean proportion of $\mathcal{C}_m$ in the $k$-hop neighbors of nodes from $\mathcal{C}_l$.
>
> **Q3. There lack insightful discussions about Proposition 3.1.  What are the physical meanings of $g(\Pi)$ and $g[\Pi]$? What does Proposition 3.1 tell us? And, what is the connection between Proposition 3.1 to the proposed algorithms?**
>
> In our revision, we add more discussion about proposition 3.1 and interaction probability.
> In short, in Proposition 3.1 we propose two inequalities - one is to show the symmetric self-interaction probability is the lower bound of the original self-interaction probability; another is to illustrate a useful inequality of symmetric self-interaction probability which is leveraged to derive a lower bound of our prediction error in Section 4.2. $g(\Pi)$ is a function with respect to the interaction probability matrix $\Pi$ while $g[\Pi]=R^{-1}Y^\top g( P)Y$, that is, $g$ is not applied to $\Pi$ but applied to the transition probability matrix $P$. When $Y$ is a square matrix, $g[\Pi]=g(\Pi)$, otherwise they are unequal. However, concerning that the notation of $g[\Pi]$ sounds confusing, we revise the notation as $\tilde{g}(\Pi)$ in our revision. The connection between Proposition 3.1 and the proposed algorithms is: Proposition 3.1 is directly used for our following theoretical analysis which inspires the proposed algorithms.
>
> **Q4. In Def. 3.2, what do you mean about the distributional representation? The authors may want to provide further explanation. In Proposition 3.2, "f be the frequency of signal x" --> "f be the frequency distribution of signal x"?**
>
> We revise Def.3.2 to make the definition more clear. Def.3.2 defines the frequency of a signal as a random variable taking values in the set of graph frequencies. We define the probability of the value of signal frequency based on the graph signal spectrum and such probability describes the frequency distribution of the signal. In our paper, we regard the graph signal spectrum as its spectral representation and the frequency distribution as the distributional representation of the signal. In Proposition 3.2, f is the frequency of signal x or we can also say f is the frequency distribution of signal x.
>
> **Q5. The low-pass/high-pass filters seem very similar to "Beyond Low-frequency Information in Graph Convolutional Networks". Moreover, the authors shall provide theoretical proofs to show why the g_b is a band-pass filter.**
>
> Note that our goal is not to develop a new filter for GNNs. Instead, we aim to analyze the prediction ability of classical/commonly used graph filters and find a way to improve. That is why we formulate low-pass/high-pass filters similar to "Beyond Low-frequency Information in Graph Convolutional Networks".  In fact, both the model proposed by the reference and our algorithms do not directly use the low-pass/high-pass filters directly. The reference proposed an attention mechanism to learn the attention weights of low/high-frequency information respectively, while our final model only uses a set of band-pass filters. Therefore, even though the formulations of low-pass/high-pass filters in our theoretical analysis part are similar to the reference, it will not diminish our novelty and contribution.
>
> $g_b$ is a band-pass filter:
> We define the band-pass filter as a quadratic function with respect to the adjacency matrix $g_{b(\epsilon')}(\tilde{A}) = I-(1+|\epsilon'|)^{-2}(\epsilon' I-\tilde{A})^2=(1+|\epsilon'|)^{-2}((1+|\epsilon’|-\epsilon’)I+A)((1+|\epsilon’|+\epsilon’)I-A)$ which is exactly an overlap between a low-pass filter $(1+|\epsilon’|-\epsilon’)I+A)$ and a high-pass filter $((1+|\epsilon’|+\epsilon’)I-A)$. Thus, $g_b(\epsilon’)$ is exactly a band-pass filter.

---

> ### Author Response · Authors · 2021-11-23
> **Response to Reviewer RH1y - Part II**
>
> **Q6. The authors claimed that this paper provides a "data-driven" mechanism for filter bank selection, which is not well described in the algorithm description.**
>
> In our algorithm, although the form of our band-pass filter  $g_b(\epsilon)$ is predefined, its parameters including $\epsilon$ and weight $\omega$ are learned from specified graphs. Moreover, the parameters of our feature disentanglement blocks (the linear transformations and learnable masking) are also learned from data which will affect the learning of the filter bank. Therefore, our filter bank selection is data-driven.
>
> **Q7. The theoretical analysis and algorithm feel disconnected.**
>
> We admit that not all the model choice is directly from the theoretical analysis but in general, the gap is small. One of the main conclusions of our theoretical analysis is that a single filter is not enough for graphs with diverse self-interaction probabilities (please refer to the last paragraph of Section 4.1 and Section 4.2). According to our analysis, as long as we add up enough band-pass filters, even if the spectra of labels are highly dispersive, we can capture all useful frequency information. So, using a filter bank to design our model is a natural choice.
>
> However, it is expensive and may introduce noise of node features if we use a large number of band-pass filters together. Concerning this, we propose two simple algorithms attempting to learn filter banks more efficiently and effectively. Although our algorithms (especially the disentanglement part) are not directly derived from our theoretical analysis, the frequency distribution we defined in the theoretical analysis part helps us validate whether the graph filters we learned can capture useful frequency information (Figure 2). In our revised version, we added more explanations of the motivations for model design.

---

### Author Response · Authors · 2021-11-23
**Paper revision**

As per the reviewers’ comments, we make some revisions to the paper and mark them in blue. To summarize, the main revisions are as follows:

1. Simplified Notations in Section 3.1 and more discussion about Proposition 3.1;

2. We added more explanations for the motivation of disentanglement in Appendix A.3;

3. We added extra experimental results in Appendix A.4 to show the advantage of using disentanglement by comparing it with a new baseline GPRGNN.

---

### Decision · Program_Chairs · 2022-01-20

**Decision:**

Reject

**Comment:**

In this paper, in order to theoretically investigate the relationship between graph structure and labels in GNNs, interaction probabity and frequency indicators are introduced and analyzed, and a new family of GNNs with multiple filters is proposed based on the insights from the theoretical analysis,
In the discussion, there was an opinion that the theoretical analysis is interesting, but its novelty and clarity are limited. Although certain contributions are acknowledged, the impact is marginal and the audience for which this paper will matter is rather limited.